# Spectral field mapping in plasmonic nanostructures with nanometer resolution

J. Krehl[1], G. Guzzinati [2], J. Schultz[1], P. Potapov[1], D. Pohl[1,3], Jérôme Martin[4], J. Verbeeck[2], A. Fery[5], B. Büchner[1] & A. Lubk[1]

Plasmonic nanostructures and -devices are rapidly transforming light manipulation technology by allowing to modify and enhance optical fields on sub-wavelength scales. Advances in this field rely heavily on the development of new characterization methods for the fundamental nanoscale interactions. However, the direct and quantitative mapping of transient electric and magnetic fields characterizing the plasmonic coupling has been proven elusive to date. Here we demonstrate how to directly measure the inelastic momentum transfer of surface plasmon modes via the energy-loss filtered deflection of a focused electron beam in a transmission electron microscope. By scanning the beam over the sample we obtain a spatially and spectrally resolved deflection map and we further show how this deflection is related quantitatively to the spectral component of the induced electric and magnetic fields pertaining to the mode. In some regards this technique is an extension to the established differential phase contrast into the dynamic regime.

[1] IFW Dresden, Helmholtzstr. 20, 01069 Dresden, Germany. [2] EMAT, University of Antwerp, Groenenborgerlaan 171, 2020 Antwerp, Belgium. [3] Dresden Center for Nanoanalysis, TU Dresden, 01062 Dresden, Germany. [4] Institut Charles Delaunay – Laboratoire de nanotechnologies et d'instrumentation optique, UMR CNRS 6281, Université de Technologie de Troyes, 10010 Troyes, France. [5] IPF Dresden, Hohe Str. 3, 01069 Dresden, Germany. Correspondence and requests for materials should be addressed to J.K. (email: j.krehl@ifw-dresden.de) or to A.L. (email: a.lubk@ifw-dresden.de)

Surface plasmon resonances (SPR) including surface plasmon polaritons (SPP) are delocalized electron oscillations that can be excited in the spatially confined electron gas on the surface of metallic nanostructures. SPRs are characterized by very intense (up to MV/m) and localized (down to nanometers) transient electrical fields, which are strongly sensitive to the environment of the nanoparticle and can be excited by external optical fields. Both the field enhancement and the confinement make SPRs attractive for the sub-wavelength control of electromagnetic fields in the infrared to ultraviolet range, with potential applications ranging from the miniaturization of conventional radiofrequency devices[1] to the realization of novel electronic, so-called plasmonic, devices, providing optical logic circuits on the nanoscale. The progress in this field is steady, and many applications have been demonstrated, such as on-chip light spectrometers and linear accelerators[2,3], plasmonic rectennas for the harvesting of light[4], enhanced Raman spectrometers[5], or LEDs and photovoltaics with a higher efficiency[6]. Moreover, metamaterials have been designed to exhibit exotic properties, such as negative refractive index and slow light propagation[7,8], as well as flat metalenses[9].

The advancement of plasmonics and related techniques relies on a thorough understanding of the properties of plasmonic resonances, such as the spatial distribution of the transient electromagnetic fields associated to different resonance modes. In the typical examples of metallic nanoparticles or nanostructures, the surface of which sustains several excitations within the low-dissipative regime, which posses different spatial distributions of the induced (surface) charges and fields. While different modes often posses different energies, energy degeneracy is possible and common in highly symmetric systems (such as nanoprisms or long chains of nanoparticles).

Optical far-field spectroscopies allow to probe only those modes, which posses a non-vanishing dipole moment (referred to as bright modes), down to mesoscopic length scales. Near-field optical spectroscopies, e.g., based on scanning near-field optical microscopy (SNOM)[10] or photoelectron emission microscopy (PEEM)[11], are surface sensitive techniques, which allow to enhance resolution down to 10 nm and to see the dark modes. Electron energy loss spectroscopy (EELS) in the transmission electron microscope (TEM) even permits to image SPRs down to nanometer resolution[12]. Here, an evanescent external field is produced by the focused electron beam that is scanned over the sample. This induces a charge separation in the metallic nanostructure, which in turn acts on the beam electrons. In essence the electron beam interacts with itself via the retarded plasmonic response. Accordingly, the Cartesian component of the induced electric field parallel to the electron beam direction (named $z$ in the following) causes an energy loss, whereas the remaining two components deflect the electrons from the optical axis. However, the spatially resolved EELS spectra (referred to as spectrum images) recorded with scanning TEM (STEM), only records the energy loss and hence only probes the $z$-component of the induced electrical field. Therefore crucial properties of the SPRs, such as the electric field enhancement in lateral directions (required for the characterization of the optical coupling) or the differentiation of modes with similar $z$-fields but different lateral behavior, are currently not experimentally detectable and may only be indirectly inferred from simulations (e.g., ref.[13]). Furthermore, the missing lateral components prevent a direct tomographic reconstruction of the dielectric response, which is therefore currently only possible in (highly symmetric) systems exhibiting only a small number of modes, which may be inferred from only a small number of projections[14,15]. Similar restrictions pertain to other high spatial resolution plasmon mapping techniques; most notably SNOM allows to map the photonic local density of states (LDOS) in the vicinity of surfaces, which also constitutes a subset of the complete plasmonic response only[16].

In the following, we present a novel approach that allows the reconstruction of the lateral components of the induced fields by probing the lateral deflection of the inelastically scattered fast electrons, referred to as inelastic momentum transfer (IMT) measurement in the following. Our approach is a generalization of the elastic center of mass (CoM) or differential phase contrast (DPC) technique widely employed to probe static electric and magnetic fields on the nanoscale[17–19].

## Results

**Theory.** In our experiment, a focused electron beam with a diameter of some tens of nanometers is scanned over and around a metallic nanostructure, where it interacts inelastically with the SPR. After the interaction the electron beam passes through the TEM's projection system, which forms a far-field diffraction pattern. This diffraction pattern is then imaged through a magnetic prism into an energy-dispersive plane, where a slit selects a chosen energy range. Subsequent optics are used to reform the diffraction pattern with the energy-filtered electrons on a pixelated detector (pixel coordinates $\boldsymbol{k}_\perp$ corresponding to lateral momentum, see Fig. 1). The recorded dataset consists of a whole map of energy-filtered diffraction patterns (EFDP) in dependence of the probe position and the selected energy $\hbar\omega$ (slit position and width), obtained by scanning the probe over the sample. This setup is similar to established techniques such as elastic CoM or DPC[18,19], with the added step of energy filtering.

The conventional energy loss signal can then be calculated by integrating the intensity of the EFDP, i.e.,

$$\Gamma(\omega) = \frac{\int d^2 k_\perp\, I(\boldsymbol{k}_\perp, \omega)}{(I_0 \Delta\omega)}, \tag{1}$$

whereas the mean deflection or lateral momentum is obtained by computing the CoM of the EFDP, i.e. $\boldsymbol{p}_\perp(\omega) = \hbar \int d^2 k_\perp\, (I(\boldsymbol{k}_\perp, \omega)\boldsymbol{k}_\perp)/\int d^2 k_\perp\, I(\boldsymbol{k}_\perp, \omega)$. Both quantities may be related to the transient electric and magnetic fields associated to the plasmonic response within the conventional semiclassical approximation (SCA). In this approximation spectral densities for the energy loss $\Gamma(\boldsymbol{r}_{0\perp}, \omega)$ and the beam deflection $\boldsymbol{p}_\perp(\boldsymbol{r}_{0\perp}, \omega)$

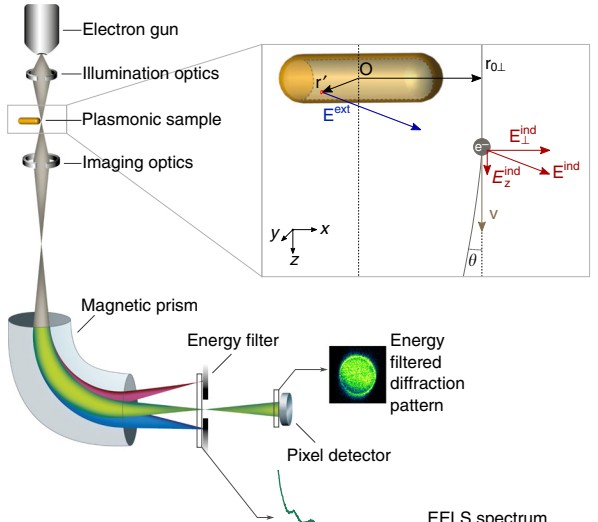

**Fig. 1** IMT setup. By using an energy filter, the angular distribution of the inelastically scattered electrons in the far field is recorded for a fixed energy loss. The CoM of that signal corresponds to the IMT, whereas the integral represents energy-loss probability for that energy loss

are obtained from classical path integrals (parameterized by the impact parameter $r_{0\perp}$, i.e., the lateral electron beam position in the object plane), which is largely valid for the deflection of fast electrons. For the sake of simplicity we work in the non-retarded approximation (i.e., we neglect the retardation of induced fields[20]) in the following derivations (see Supplementary Note 2 for more details). Moreover, we consider a single ray of electrons moving with velocity $|\mathbf{v}| = v_z$ along the optical axis $z$ (i.e., $\mathbf{r}_\perp(t) = \mathbf{r}_\perp$ and $z(t) = v_z t + z_0$) and use the no-recoil approximation (NRA, electron deflection neglected for computing induced field) taking into account that inelastic scattering angles are very small. Using that framework, the experimentally measured spectral density of the energy loss $\Gamma(r_{0\perp}, \omega = \Delta E/\hbar)$ may be expressed in terms of the induced transient electric field along the optical axis, viz.:

$$\Gamma(\mathbf{r}_{0\perp}, \omega) = -\frac{e}{\pi\hbar\omega} \int_{-\infty}^{\infty} dz\, \Re\left\{ e^{-i\omega z/v_z} \widetilde{E}_z^{\mathrm{ind}}(\mathbf{r}_{0\perp}, z, \omega) \right\}. \quad (2)$$

Here, $\Re$ takes the real part of the argument including the spectrally resolved electric field in $z$-direction, which is sufficient because classical fields in the time domain are real quantities. Using the same semiclassical reasoning and transformation steps, the expectation value of the IMT reads

$$\langle \mathbf{p}_\perp(\mathbf{r}_{0\perp}) \rangle = \int_0^\infty d\omega\, \Gamma(\omega) \mathbf{p}_\perp(\mathbf{r}_{0\perp}, \omega) \quad (3)$$

yielding the following relationship between the IMT and the induced lateral fields:

$$\begin{aligned}\mathbf{p}_\perp(\mathbf{r}_{0\perp}, \omega) = &-\frac{e}{\pi v_z} \int_{-\infty}^{\infty} dz\, \Re\left\{ \frac{e^{-i\omega z/v_z}}{\Gamma(\mathbf{r}_{0\perp}, \omega)} \widetilde{\mathbf{E}}_\perp^{\mathrm{ind}}(\mathbf{r}_{0\perp}, z, \omega) \right\} \\ &-\frac{e}{\pi} \int_{-\infty}^{\infty} dz\, \Re\left\{ \frac{e^{-i\omega z/v_z}}{\Gamma(\mathbf{r}_{0\perp}, \omega)} \left( \left[ -\widetilde{B}_y^{\mathrm{ind}}, \widetilde{B}_x^{\mathrm{ind}} \right]^T (\mathbf{r}_{0\perp}, z, \omega) \right) \right\}. \end{aligned} \quad (4)$$

Accordingly, the IMT also contains a (small) contribution from induced magnetic fields in contrast to the energy-loss probability only depending on induced electric fields in beam direction. We may additionally introduce the notion of the dyadic Green's function $\mathbf{G}$ and the dielectric susceptibility tensor $\chi$, which accounts for the dielectric response of the nanostructure towards external currents according to $\mathbf{E}^{\mathrm{ind}}(\mathbf{r}, \omega) = -4\pi i\omega \int d^3 r'\, \mathbf{G}(\mathbf{r}, \mathbf{r}', \omega) \mathbf{j}^{\mathrm{ext}}(\mathbf{r}', \omega)$ and external fields according to $\widetilde{\mathbf{E}}_\perp^{\mathrm{ind}}(\mathbf{r}_{0\perp}, z, \omega) = \int d^3 r' \left[ \chi(\mathbf{r}, \mathbf{r}', \omega) \widetilde{\mathbf{E}}^{\mathrm{ext}}(\mathbf{r}', \omega) \right]_z$, respectively, which in our case are produced by the impinging electron. Accordingly, only the $z$· components of the above two tensor fields are probed in conventional EELS, whereas the $x$· and $y$· components contribute to the IMT signal as explored in this paper.

As a consequence of the scanned probe only measuring fields it itself induced, both energy loss spectroscopy and IMT are not sensitive to the phases of the oscillating surface plasmon mode, if measured with an ideally focused STEM probe. Using an extended probe, the non-local response, including phase effects, affects the recorded signal. The above relations may be generalized to this case by employing the (generalized) Ehrenfest theorem (see Supplementary Note 2). These and other generalizations (e.g., inclusion of retardation, magnetic fields) have been used to compute both the energy loss, as well as the deflection of the real STEM probe with existing semiclassical methods commonly used to simulate SPRs (see Methods). More specifically, we employed a numerical code based on the boundary element method (BEM) approach to compute the spectrally resolved dielectric response, i.e. the induced electric fields, pertaining to the SPR occurring in our metallic nanostructure[21–23].

**Experimental results.** In the following, we demonstrate spectral field mapping at a surface plasmon mode of a lithographically

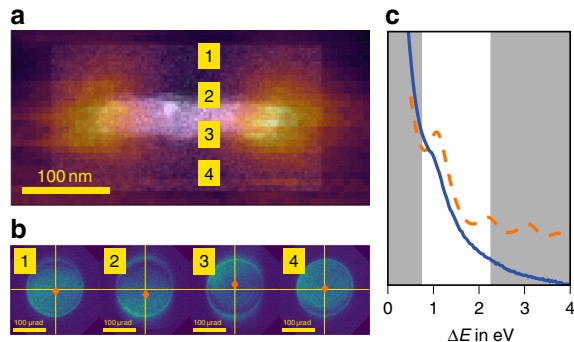

**Fig. 2** EELS and IMT measurements. **a** HAADF image (black and white) and overall loss probability (color overlay), **b** subset of inelastic EFDPs with indicated centers-of-mass, and **c** experimental (solid line) and simulated (dashed line) overall EEL spectra with employed energy slit indicated. The beam positions are indicated with respect to the Al rod. Note that an effective dielectric screening (see Methods), accounting for the aluminum oxide surface and the substrate, was necessary to shift the simulated peak to the experimentally observed value. The strong increase of the EEL spectra towards smaller losses are due to the monopole surface mode centered at approximately 0.7 eV

produced Al nanorod, which was probed with an electron beam accelerated with 120 kV (see Methods for the details of the setup). We employed a particularly large camera length to resolve small changes in scattering angle and selected the most prominent SP mode, namely the dipole mode along the long axis of the Al rod (Fig. 2a). Note the positive sign of the loss probability of the dipole mode at both caps reflecting the previously discussed insensitivity toward phases of the oscillating SP within the STEM setup. We recorded two sets of EFDPs under the same imaging conditions, one within an energy-loss window ($E_{\mathrm{Al}} = 1.5 \pm 0.75$ eV, see Fig. 2c) containing the desired dipole mode, and a second one within $0 \pm 0.75$ eV, containing the elastic diffraction patterns, which are necessary in order to evaluate and eventually correct for elastic scattering effects, such as vignetting or charging (see Supplementary Notes 1 and 3). Accordingly, we observe variations and beam deflections to be smaller by a rough factor of two in the elastic EFDP compared to the inelastic for our scenario. Therefore, we can consider a significant part of the DP variations in this scenario to be of exclusively inelastic origin (see Supplementary Note 4 for a different scenario using Au spheres).

The observed modifications in the EFDP (Fig. 2b) consist of both, an intensity variation within the EFDP, as well as a displacement of the diffraction disks, with the largest variation showing up when the beam is placed on the particle boundary. The shift of the whole disk represents the inelastic momentum shift, whereas the redistribution of intensity within the disk may be also related to elastic and inelastic vignetting. In an extended convergent probe the angle of incidence changes over the position inside the probe, so partial obstruction in real space leads to a shift in the far field (i.e., a change in the mean lateral momentum, see the Supplementary Information). This vignetting effect can be caused by elastic scattering absorption in the particle or the sharp screening of the loss probability of the surface plasmon inside the specimen. The impact of vignetting may be suppressed by either reducing the convergence angle (as done in our example) or ultimately removed by deconvolving the initial phase space distribution of the probe from the EFDP as discussed further below.

Figure 3 contains a summary of the results obtained as well as simulations for comparison. By normalizing the integrated intensities of the EFDPs according to Eq. (1), we can

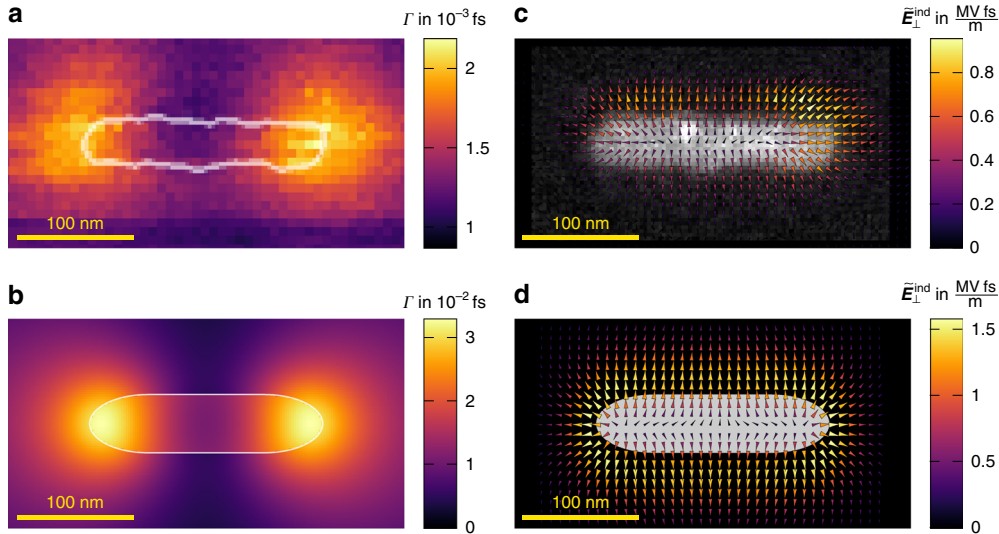

**Fig. 3** Comparison of reconstructed and simulated quantities. Energy-filtered experimental (**a**) and simulated (**b**) energy loss probabilities (Eq. (2)) within the energy interval indicated in the spectra (Fig. 2b) and the experimental (**c**) and simulated (**d**) electrical field maps (Eq. (4)) of the Al nanorod. The field maps show the transient field for the specific energy loss (i.e. a spectral component) selected by the slit

quantitatively map the energy-loss probability, which corresponds to the spatial distribution of a certain projection of the spectrally resolved $\tilde{E}_z$ field (Fig. 3a, b) following Eq. (2). Note that this is the same type of signal obtained from conventional spatially resolved EELS. The generally good qualitative agreement between experiment and simulation in these maps (Fig. 3a, b) proves the high stability of EFDP imaging conditions, although a decrease of loss probability toward the lower part of the map due to some remaining energy drift during long acquisition times could not be completely avoided. Moreover, there is a noticeable contribution from the monopole mode at lower energies, producing significant loss probability also on the long side of the rod, where the dipole mode drops to zero. Note, however, the large mismatch in magnitude comprising approximately one order of magnitude. We attribute this difference to persistent shortcomings of the effective dielectric screening approach, insufficiently incorporating the attenuation of the induced fields due to the oxide and carbon surface layers as well as the substrate.

Going one step further, we may now map spectral resolved lateral electric fields by normalizing the experimentally observed projected fields with the particle thickness (magnetic fields contribute roughly 10% to the overall deflection and are neglected in the following, see Methods). Taking into account the energy drift towards the lower edge of the scan, we obtain average spectral field strengths of approximately MV fs m$^{-1}$ at the particle surface, in good agreement with the simulations (Fig. 3c, d). The normalization with the particle thickness is justified by the confinement of the lateral fields to the particle geometry, as evidenced by the simulated 3D representation of the transient fields in Fig. 4 for this energy loss. The distribution of the lateral field of the dipole mode (Fig. 3c, d) is more homogeneous with a slight pronunciation at the particle caps only. Beside the energy drift-induced attenuation toward the lower edge, it shows some additional deviations, which we attribute to the irregular surface of the Al rod not taken into account in the simulations.

The 3D representations in Fig. 4 provide an intuitive picture for inelastic electron scattering and indicate possible interpretations of the energy-loss probability and IMT in terms of fields. Accordingly, the inelastically scattered electrons are deflected in direction of the respective closest surfaces and the magnitude of the deflection quickly decays with the distance to the surface. This

may be explained qualitatively by an induced mirror charge, attracting the inelastically scattered electron. In accordance with that picture, the field is predominately radial around the specimen, which implies that the $z$-component is almost antisymmetric with respect to the middle plane ($z = 0$) of the Al rod. When the $z$-component is integrated along the beam direction, the dominant antisymmetric component vanishes and only the small symmetric part contributes to the energy loss signal. The lateral field components, however, are chiefly symmetric in $z$ and appear therefore largely undiminished in the IMT. In our measurement, and corroborated by simulations, the projected field of the $z$-component is smaller by roughly one order of magnitude compared to the lateral components. In other words the lateral components permit a direct quantitative interpretation in terms of average fields, whereas the link between $z$-component and energy loss is indirect and more susceptible to measurement errors, as well as the details of the particle environment (e.g., substrate). We partly attribute the disagreement in total magnitude between simulated and experimentally observed energy loss probability to the latter effect.

## Discussion
In summary, we have demonstrated the quantitative mapping of spectrally resolved induced electric fields employing the IMT technique in the (S)TEM. The results reveal the projection of all components of the transient fields pertaining to a selected mode and by extent the dielectric susceptibility tensor, which determine the optical properties of plasmonic nanostructures. This method paves the way for further generalizations comprehensively probing the plasmonic responses with nanometer resolution in the TEM. Firstly, analyzing the complete energy filtered diffraction patterns (instead of its first two moments) via energy-filtered ptychography[24] yields the full non-local dielectric response of a system (see Supplementary Note 1). This enables for instance the characterization of quantum effects (e.g., Lindhard screening or tunneling in strong plasmonic fields), the transport behavior of surface plasmons in complex nanostructures or interfaces thereof or the influence of the crystal field on plasmons. Ptychography also allows the separation and removal of source shape and vignetting due to elastic scattering in the sample, thereby increasing

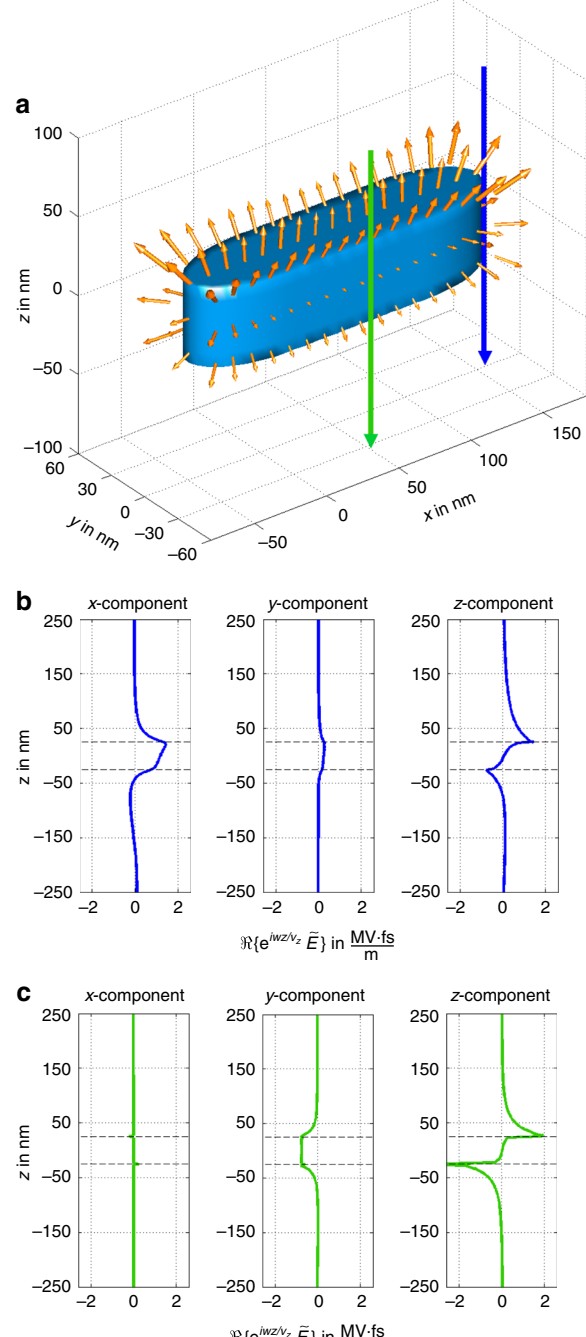

**Fig. 4** Simulations of induced fields. Simulated spectrally resolved electric field distribution in **a** 3D and **b**, **c** 1D along lines indicated in **a**. The fields exhibit a strong confinement to the rod surface as well as the symmetry (antisymmetry) of the $x$, $y$ ($z$) components with respect to the central plane ($z = 0$) of the rod. Consequently, representative values of the lateral fields may be obtained from the 2D projections after dividing with the rod thickness. Please note that the integral of the $z$-component, as in **b**, **c**, is always positive due to the slower decay of the field above the sample

the spatial resolution of the reconstructed response. Secondly, a combination of IMT with multiple tilt-axis tomography (i.e., tensor field tomography) would allow the 3D reconstruction of all components of the dielectric response without relying on additional assumptions. This relates to and includes the so-called optical density of states (i.e., the spatial diagonal of the dyadic Green's function $G(r, r' = r, \omega)$ mentioned in Theory), which has

been previously reconstructed within model-based approaches[15]. Thirdly, the IMT contains information about deflecting magnetic fields and hence surface currents, which may be exploited to characterize so-called magnetic modes[25], i.e., peculiar surface plasmonic modes including surface ring currents. Last but not least, one may synthesize the time-dependent dielectric response from the IMT measurements at a whole range of energy losses, providing access to time-resolved fields complementary to that of ultrafast TEM techniques[26,27]. These extensions may be used to reveal crucial parameters, such as the optical density of states, facilitating the investigation of the spatial variations of the dielectric response due to, e.g., chemical inhomogeneities, geometric variations, as well as non-local quantum effects beyond the classical electromagnetic response with unprecedented resolution.

## Methods

**Sample production.** The aluminum structures have been fabricated using electron beam lithography (EBL) in a FEG SEM system (Raith eLine). First, a 150 nm thick layer of poly(methyl methacrylate) (PMMA) was spin-coated on a STEM-EELS compatible substrate. This consists of arrays of 15 nm thick $Si_3N_4$ square membranes engraved in a small silicon wafer (3 mm diameter). The membranes were subsequently impressed by the electron beam using the EBL system (doses varying between 150 and 300 μC/cm²). The patterns in the resist were then developed for 60 s in a 1:3 MIBK:IPA solution at room temperature. Then a 40 nm thick layer of Al was deposited on the sample using thermal evaporation (Plassys ME300). Finally, the lift-off has been performed by immersing the sample in acetone unveiling the Al structures on the membranes. The width of the nanostructures is between 40 and 50 nm and they are of an approximately uniform thickness of 50 nm. Finally, the Al rods have been coated with a thin carbon layer to reduce charging.

**TEM experiments.** Experiments have been carried out at an FEI TITAN³ TEM (acceleration voltage $U_{acc} = 120$ kV) equipped with a Wien-type monochromator and an energy filter (Gatan Quantum). While an EFDP is easily recorded by projecting the diffraction pattern inside a conventional post-column image filter, the accurate measurement of very small deflection angles (tens of rad) requires the careful optimization of the experiment parameters. The semi-convergence angle, as well as the angular pixel-size of the detector, must be greatly reduced with respect to a conventional setup. By almost completely switching off the objective lens of the microscope, we were able to achieve a semi-convergence angle of 170 μrad and an angular resolution of of 1.24 μrad/pixel. The beam was then scanned in a raster fashion over a region of 64 by 32 points, acquiring a (energy filtered) diffraction pattern of 256 px by 256 px (with a 8-fold binned detector) for each beam position. The step size for this raster scanning (approximately 3.5 nm) was intentionally chosen to be lower than the beam's diameter 20 nm in anticipation of using the dataset for ptychographic analysis. The dwell time was 0.3 s at a current of ≈0.1 pA.

**Data treatment.** To evaluate the IMT from the EFDP, the drift of the beam over the detector during the scan was compensated by estimating an offset-wedge from the outermost pixels (whose CoM should be about zero) and shifting the diffraction patterns with these offsets. Afterwards the rotation between the scanning dimensions and detector coordinates was compensated, from a calibration measurement using a reference specimen. Note, however, that small drifts of the filtered energy window due to instabilities in the filter could not be corrected a posteriori, leading to small errors in the mapped fields. Subsequently, the CoM has been evaluated and normalized to the total beam fluence. To account for the averaging over the selected energy range, the scanning position dependent peak-to-mean ratio of the loss probability was estimated from a previously recorded spectrum image and applied to the extracted datasets; the correction ranged from 1.3 to 1.6.

**Numerical simulations.** The numerical simulations of the plasmonic response of the nanostructure have been performed with the Matlab-based open-source toolbox MNPBEM[21] with dielectric function for Al taken from the Drude model (using a plasma frequency $\omega_P = 15.826$ eV and a damping constant of $\gamma_D = 1.606$ fs). To approximate the impact of the dielectric environment formed by the ubiquitous $AlO_2$ oxide layer, the carbon layer and the $Si_3N_4$ substrate on the plasmonic response, we increased the dielectric constant of the surrounding medium to 3.5 by aligning the mode in the experimental and simulated loss spectra (there was, however, little change above 2.4 in the loss probability and the lateral momentum transfer). This effective screening has been used in previous studies[28,29] and introduces a significant red shift on the peak position of the dipole mode and a damping of the lateral electric fields in good agreement with the experiment. Accordingly, for each pixel position the retarded response of the particle was computed using a BEM approach, yielding the spectrally resolved induced electric field components $\tilde{E}^{ind}(r_{0\perp}, z, \omega)$. Subsequently, we computed the energy loss probabilities according to Eq. (2) and the IMT according to Eq. (4). Moreover, we

computed both the non-retarded and the full solutions in order to assess the impact of retardation and magnetic fields. We found that the magnetic field contributed a fraction in the range of 10% to the overall beam deflection.

## Data availability

The experimental raw data used in this word are available from the corresponding authors upon request.

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

## Acknowledgements

G.G. acknowledges support from a postdoctoral fellowship grant from the Fonds Wetenschappelijk Onderzoke-Vlaanderen (FWO). A.L. and J.K. have received funding from the European Research Council (ERC) under the Horizon 2020 research and innovation program of the European Union (grant agreement no. 715620).

## Author contributions

J.K. performed the data treatment and IMT reconstruction. J.S. conducted the BEM calculations. G.G. recored the Al EFDP. D.P., P.P., and A.L. recorded supplementary Au EFDPs. A.L. provided the idea and the theory for the experiment. J.M. and A.F. prepared samples. B.B. supervised the project. G.G., J.K., P.P., A.L., and J.V. wrote the manuscript.

## Additional information

**Competing interests:** The authors declare no competing interests.

