## [Peer Review File · Nature Communications]

Reviewers' comments:

Reviewer #1 (Remarks to the Author):

The authors employ a variant of electron energy loss spectroscopy to measure the deflection of an electron beam when raster-scanned over a plasmonic nanoparticle, and estimate from the results the forces exerted by the plasmon fields. Quite generally, the topic is timely and the technique could provide information that, to the best of my knowledge, is very hard to obtain otherwise. However, I do have a number of difficulties with the paper in its present form.

Presentation.

At several places more information should be provided. For instance, at the beginning of the discussion of the experimental results the authors talk about two datasets, but provide practically no details on how the second set was obtained. In Fig. 2c it is not indicated which curve corresponds to experiment and theory. It also appears that some ZLP contribution is added to the simulation results. No information is given about the parameters of the Al Drude function. I also think that the abstract should be strengthened, "we show that this deflection may be related to the induced fields" is not a very strong statement.

Theory.

In the theory section the authors extensively refer to the supplementary information, which, however, is very sketchy, technical, and hard to read. As far as I understand, the authors' approach is based on Ritchie's theory for computing EELS probabilities. This approach starts with a quantum description where the electron wavefunction is decomposed into plane waves, and then shows that the loss probabilities can be reduced to a semiclassical expression (using classical electron trajectories) when the spectrometer records all plane waves. Looking to Fig. 2b it seems that this is not the case here. Can the authors elaborate on this and provide more justification for their semiclassical approach? I would also like to see a more detailed discussion of what could be extracted from the full inelastic EFDP maps and not just the center positions.

Nanoparticles.

My main problem with the paper, however, is the use of Al rods and of missing reference measurements. For reasons that are not clear to me the authors decide to use Al nanoparticles which suffer from strong damping, rather than using conventional gold or silver nanoparticles. The authors state that they selected the most prominent SPR peak, from Fig. 2c it appears that even this peak is extremely faint. Even worse, Al is known to possess an interband transition at approximately the same energy, see for instance the data of Palik, so one may wonder whether this peak has a plasmonic origin at all. Along the same lines, I find the background dielectric constant of 3.5 needed in the simulations to reproduce the experimental results too high, it seems that something strange is happening here. Additionally, for a conceptual paper that introduces a novel technique I would like to see at least one additional nanoparticle geometry, ideally two coupled rods with strong fields in the hot-spot region, in order to estimate whether the experimental results coincide with the expectations for such a geometry.

For the above reasons, I think that the present work is not mature enough yet, and I recommend against publication in Nature Communication. I believe, however, that this is an interesting technique with a lot of potential. If the authors could manage to perform additional experiments with plasmonic nanoparticles having more prominent SPR modes, ideally made of gold or silver, I would recommend reconsideration of the paper in Nature Communications, otherwise I suggest resubmission to a more specialized journal.

Reviewer #2 (Remarks to the Author):

The manuscript describes a very interesting experiment, in which both the electron energy loss (EEL) and the inelastic momentum transfer (IMT) of an electron beam passing an Al nanorod is shown. Specifically, the energy loss probability, a frequently investigated quantity, is combined in a novel way with the deflection of the electron beam, in order to obtain both longitudinal and transverse components of the electron's self-induced field. While the spectrally resolved loss probability is a measure of the longitudinal induced field, the deflection measures the respective spatial Fourier component of the transverse field(s). In the analysis, a straightforward and well-justified theoretical description is given, with expressions linking the measured deflection with the position-dependent induced transverse field.

I believe this is the first experiment analyzing energy-filtered deflection maps for a resonant nanostructure. The experimental approach is innovative, the results are generally convincing and of good quality, and I generally believe the manuscript has potential for publication in Nature Communications. However, there are some fundamental issues concerning the presentation and placing into context of the manuscript, which need careful attention by the authors to avoid misconceptions, before the manuscript could be further considered.

-Specifically (and I am not suggesting that they are doing so willingly), the authors appear to blur the lines between (1) a map of the induced field on the electron trajectory as a function of scan position near a nanostructure and (2) a map of a plasmonic mode of that nanostructure. In particular, several phrases in the text, the title, abstract and the quiver plot in Figures 3 and 4 appear to give that very impression. Indeed, one of the main claims of the manuscript is that it achieves a mapping of plasmonic fields, specifically plasmonic modes. Expressedly, the manuscript states: "In the following, we demonstrate the spectral field mapping of a surface plasmon mode of a lithographically produced Al nanorod, which was probed with an electron beam accelerated with 120 kV (see methods). For that, we selected the most prominent SPR mode, namely the dipole mode along the long axis of the Al rod." These are rather problematic claims, and from my perspective they are not sufficiently substantiated. These statements would imply to most readers that the field distributions obtained from the deflection maps, corresponding to the induced field, were closely related in direction and strength to the plasmon mode studied. However, it is evident that the induced field as a function of scan position will consistently point towards the nanostructure, which will always yield induced-field distributions with the same symmetry as the investigated structure, i.e., lacking a dipole contribution. This, of course, does not in any way reflect the symmetry of the plasmon dipole mode, which, as excitable by far-field radiation, is antisymmetric in nature. In other words, the technique is unable to obtain overall position-dependent phase information (unless other techniques, see below). Conversely, it is hard to see how a field component tangential to the structure (parallel to the rod axis) could be mapped, which is present in the plasmonic mode, but does not correspond to the structural symmetry. Overall, I am rather certain that readers from plasmonics will be confused when they see the monopolar (+quadrupolar) type of "field maps" in Figure 3.

-The authors frequently use the term "transient field". For example, the caption in Fig. 3 calls these arrow plots "transient electrical field maps of the Al nanorod". While the field is certainly transient for an electron passing the nanostructure, I believe most readers will expect the measurement of "transient fields" as being related to some time-resolved measurement or a time-dependent phenomenon.

-The title "Spectral Field Mapping in Plasmonic Nanostructures with Nanometer Resolution" also implies that the field distributions shown relate to the plasmonic modes of the structure investigated. Also the term "Spectral Field Mapping" implies a spectrally dependent measurement, which appears possible for the future, but was not carried out.

In summary, these very nice measurements will be interesting to the plasmonic and electron

microscopy communities in themselves, and I would suggest that less emphasis is placed on those claims the paper cannot satisfy.

Additional minor points:

-The caption in Fig. 2 does not denote the solid blue and dashed lines.

-The abstract states: "Notwithstanding, the direct imaging of transient fields permitting a direct mapping of plasmonic coupling, e.g., in terms of field enhancement, has been proven elusive to date. Here, we fill that gap..."

There are notable other techniques, such as NSOM or PEEM, which provide phase-dependent field measurements in a manner at least as direct as what is shown here, I would say. This aspect is probably a little overstated in the manuscript, and corresponding references should be included.

Reviewer #3 (Remarks to the Author):

The authors present a novel method for mapping the in-plane components of the electromagnetic field in plasmonic nanoparticles as a function of energy-transfer (i.e., for selected plasmonic modes). This is achieved using a form of energy-filtered differential phase contrast referred to as IMT in the text. Theoretical background, experimental evidence and conclusions are all provided in a clear and convincing way and are original. I strongly believe that this work will have a significant impact on the thinking within the field of electron microscopy and also inspire scientists in other fields. Therefore, I strongly recommend publishing it in Nature Communications, provided that the changes and comments listed below are taken into account.

== Content questions/comments ==

* In the Introduction, the authors state that TEM/EELS allow to map SPRs with nanometer resolution, yet in the Theory and the Methods section, they give the beam diameter as 20 nm. In the Methods section, the authors also state that the step size was purposefully chosen much smaller than the beam diameter. Please clarify how all this affects spatial resolution (and what the purpose for choosing a small step size was).

* p. 3, left column, first paragraph: Ehrenfest's theorem has already been employed in [16] and its impact on STEM DPC has been discussed in more detail in Müller-Caspary et al., Ultramicroscopy 178 (2017) 62-80.

* Fig. 2: What are the dots in b) (COMs, maybe)? What are the two curves in c) (simulation and experiment? monochromated and unmonochromated?)

* In Fig. 2b, images 1 and 4 show non-uniform (and different!) intensity distributions, not just shifts. Where does this effect come from (it should not be vignetting as these positions are arguably far away from the interface) and how does it affect the analysis?

* In Experimental Results, the authors state that "the shift of the whole disk represents the inelastic momentum shift, whereas the redistribution of intensity within the disk may be also related to elastic and inelastic vignetting". In Data Treatment, they state that "To evaluate the IMT ... the CoM has been evaluated" after correcting for artifacts. Which method was used (CoM or disk shift)?

According to [16] and Müller-Caspary et al., Ultramicroscopy 178 (2017) 62, the CoM should generally be used as it is related to the (average) deflection field by the Ehrenfest theorem. Naturally, how meaningful the average deflection field actually is will depend on the ratio between the characteristic length scales of the beam (diameter) and the field's variation, i.e. one will not be able to correctly measure fields varying on a length scale much smaller than the beam diameter.

Constant fields lead to a simple shift of the diffraction disk, which can be evaluated by itself (e.g. by circle fitting) or by CoM - which should give the same results in this case. However, vignetting, partial absorption etc. will certainly "mess up" CoM calculations. They might not affect shift measurements as severely, but those are not directly interpretable in terms of the Ehrenfest theorem. Please clarify your procedure and how you overcame these issues (such as vignetting).

* On p.3, right column, the authors state that they attribute the differences between the experiments and the simulations, among other things, to "the attenuation of the induced fields due to the oxide and carbon surface layers as well as the substrate". Were those (especially the substrate) included in the simulations at all?

* Fig. 3: The difference by a factor of ~ 10 between simulations and experiments in panels a) and b) is explained in the text (by an oxide layer and substrate effects, among other things). Why does the same effect not seem to occur in panels c) and d) for the in-plane components?

* Fig. 4: the z component profiles seem to be asymmetric w.r.t. the particle. In particular, the profile in b) seems to have larger positive values (up to ~ 1.25) than negative values (down to $\sim -.75$). In contrast, the profile in c) seems to have larger negative values (down to ~ -2.25) than positive ones (up to ~ 2). As the integrals over the profiles are related to the energy loss and the sign differs, does that mean there is energy gain in one of the cases? This might just be a drawing artifact, but should definitely be checked and corrected or mentioned and explained in the text.

* On p.5, right column, the authors mention the reconstruction of the optical density of states. This has been accomplished before using tomography, e.g., in Hörl et al., Nature Communications 8 (2017) 37.

* In the Methods section, please add some additional information, including the beam current, total number of beam positions, CCD exposure time per beam position, and number of pixels per CCD image. Also, how were the $170 \mu\text{rad}$ in STEM achieved?

* In Sample Production, the authors mention that the "Al rods have been coated with a thin carbon layer to reduce charging". How was this done and how thick was the layer? In particular, was the substrate also coated, which would presumably result in a large reservoir of free charges and therefore alter or even destroy the surface resonances.

== Minor typos and formatting ==

* p. 1, last word in the left column should be "possess", not "posses"

* In the caption of fig. 2, "the beam positions ... indicated" should be "the beam positions ... are indicated"

* In the caption of fig. 4, "exhibit a the strong" should use either "a" or "the", but not both

* The references should be ordered (the introduction cites [4], followed by [2, 23], followed a bit later by [1])

* In ref [6], there is an encoding issue with one of the names

* Ref [16] lacks page numbers

Letter to the Referees for resubmission of NCOMMS-18-07042

June 8, 2018

We thank the referees for their effort, their fairness and honesty in the reviews. You raise several good points of critique and of lack of clarity, which we strive to rectify. We hope that the manuscript thus becomes more complete and understandable to a wide audience. Especially the arguments on the theoretical derivation has helped us to better understand the involved assumptions and to develop a more rigorous and elegant approach.

In the revision process we noticed the arrows in figure 3 were the wrong way around (as field components they point opposite to the deflection of the electrons, since those are charged negatively). We flipped them.

Reviewer #1 (Remarks to the Author):

The authors employ a variant of electron energy loss spectroscopy to measure the deflection of an electron beam when raster-scanned over a plasmonic nanoparticle, and estimate from the results the forces exerted by the plasmon fields. Quite generally, the topic is timely and the technique could provide information that, to the best of my knowledge, is very hard to obtain otherwise. However, I do have a number of difficulties with the paper in its present form.

Presentation.

At several places more information should be provided. For instance, at the beginning of the discussion of the experimental results the authors talk about two datasets, but provide practically no details on how the second set was obtained. In Fig. 2c it is not indicated which curve corresponds to experiment and theory. It also appears that some ZLP contribution is added to the simulation results. No information is given about the parameters of the Al Drude function. I also think that the abstract should be strengthened, "we show that this deflection may be related to the induced fields" is not a very strong statement.

- The second reference dataset containing the zero loss filtered diffraction patterns was recorded under the same imaging conditions (convergence angle, camera length, etc.) like the SPR filtered one. We added this information.
- The experimental and simulated spectrum lines in figure 2c are now indexed in the caption.

We did not add any ZLP-contribution to the simulated results. Indeed, the increase of the loss probability toward the infrared regime, belongs to the monopole mode

appearing at these energies (as stated in the figure caption now). The presence of this mode is also apparent in the energy loss map (Fig. 3a), where it contributes the loss probabilities in the middle section of the rod, where the dipole mode drops to zero (information added to the manuscript).

- We agree that the sentence was to unnecessarily unspecific, we changed it to: “[...] we show how this deflection is related quantitatively to the induced electric and magnetic fields.”.

Theory.

In the theory section the authors extensively refer to the supplementary information, which, however, is very sketchy, technical, and hard to read. As far as I understand, the authors’ approach is based on Ritchie’s theory for computing EELS probabilities. This approach starts with a quantum description where the electron wavefunction is decomposed into plane waves, and then shows that the loss probabilities can be reduced to a semiclassical expression (using classical electron trajectories) when the spectrometer records all plane waves. Looking to Fig. 2b it seems that this is not the case here. Can the authors elaborate on this and provide more justification for their semiclassical approach? I would also like to see a more detailed discussion of what could be extracted from the full inelastic EFDP maps and not just the center positions.

- We fully understand the referees concern about the theory part. It was (and might unfortunately remain) a bit sketchy and fragmentary for the following reasons. In our initial derivation we followed the common derivation as laid out, e.g., in Abajo’s review paper [1]. However, and here we are fully in line with the referee, this is not a justification for using the classical expressions. Unfortunately, Ritchie does not resolve this problem (at least as we understand his work), because he just shows how the general energy-loss probability expression reduces to an incoherent summation over the point-wise probabilities and the incoming beam intensity if we integrate over all final states. This result doesn’t say anything about whether we may use a classical expression for the point wise probabilities or when quantum effects (e.g., Lindhard screening) have to be taken into account. Moreover, we cannot employ this trick for the computation of the inelastic momentum transfer, since it only applies for the loss probability. We therefore revised the supplement with a derivation, which, to our best knowledge, is original and grasps the essential semiclassics behind our expressions. It is rooted in the (generalized) Ehrenfest theorem, which governs the dynamics of the expectation value of an operator (Hamiltonian and lateral kinetic momentum in our case). The critical point in this derivation is the assumption that the state of the focused STEM probe is invariant, when traversing the sample (no recoil approximation). This assumption is very similar to the one used for interpreting elastic (static) differential phase contrast in terms of static fields (e.g., [2]) and we think it is well justified by the small inelastic scattering angles. Nevertheless, it remains a source of concern as quantum mechanical effects, such as Lindhard screening, manifest themselves in a variation of the outgoing state with respect to the initial state. In our view, the subject of (non-local) quantum effects in surface plasmon excitations is still not adequately understood, which provides a major motivation for the development of inelastic ptychography as discussed below.
- In order to discuss the full information content of the inelastic EFDP, we have to consider the inelastic ptychography in full detail. This is in essence beyond the scope of this paper

and necessitates a whole new block of theory. Moreover, for an overview of the analytical capabilities of ptychography we simply do not know enough about the final performance we can get out of our TEMs and data processing. Nevertheless, we added a completely new section in the supplement, where the link between the inelastic EFDP, inelastic ptychography and non-local dielectric response, is described in a condensed form. Additionally, we reformulated the Summary thusly: *“Firstly, analyzing the complete energy filtered diffraction patterns (instead of its first two moments) via energy-filtered ptychography[3] yields the full non-local dielectric response of a system (see Supplementary Information). This enables for instance the characterization of quantum effects (e.g., Lindhard screening or tunneling in strong plasmonic fields), the transport behavior of surface plasmons in complex nanostructures or interfaces thereof, or the influence of the crystal field on plasmons. Ptychography also allows the separation and removal of source shape and vignetting due to elastic scattering in the sample, thereby increasing the spatial resolution of the reconstructed response.”.*

Nanoparticles.

My main problem with the paper, however, is the use of Al rods and of missing reference measurements. For reasons that are not clear to me the authors decide to use Al nanoparticles which suffer from strong damping, rather than using conventional gold or silver nanoparticles. The authors state that they selected the most prominent SPR peak, from Fig. 2c it appears that even this peak is extremely faint. Even worse, Al is known to possess an interband transition at approximately the same energy, see for instance the data of Palik, so one may wonder whether this peak has a plasmonic origin at all. Along the same lines, I find the background dielectric constant of 3.5 needed in the simulations to reproduce the experimental results too high, it seems that something strange is happening here. Additionally, for a conceptual paper that introduces a novel technique I would like to see at least one additional nanoparticle geometry, ideally two coupled rods with strong fields in the hot-spot region, in order to estimate whether the experimental results coincide with the expectations for such a geometry.

- We appreciate the concerns raised by the referee but would like to respond to several points elaborated further below. Firstly, we do not concur that aluminium structures are per se unsuited or uninteresting for plasmonics or that the mode we acquired might not be of a plasmonic origin. Secondly, we are and have been working on gold structures and were able to extract energy-filtered deflection maps. So far, however, we have not been able to extract unambiguous inelastic momentum transfer and hence fields from them, mainly due to the strong elastic scattering of Au. And thirdly, we would like to explain and back our procedure for determining the background dielectric constant in the simulation.
1. Knowing beforehand that vignetting could overshadow plasmonic signal we looked for a sample with a small scattering absorption. At the small scattering angles, we recorded, gold has a mean free-path length (pertaining to scattering absorption) of about 16nm whereas aluminium has one of about 100nm (silver has 19nm). We therefore expected aluminium to exhibit lower vignetting and thereby a clearer plasmonic IMT signal than gold (the energy filtered deflection is always a superposition of elastic and inelastic deflection). Nevertheless we also conducted the experiment at plasmonic gold nanoparticles (cf next section further below). Our measurements at the Al nanorods clearly show the increase of the loss probability at the caps, as one would expect of the dipole mode. Consequently, we clearly captured

the dipole mode. The interband transition in that energy range does not change this dipolar character of the surface plasmon. It rather provides for an additional decay channel, broadening the spectral peak [4]. Additionally, it slightly changes the real part of the dielectric function in that energy range also shifting the original plasmonic mode. To check this influence we performed simulations with experimental dielectric functions of Al containing the interband transition [5]. However, no significant shift of the SP peak with respect to the Drude model could be observed. With that we do not see no reason that our measurements could stem from something other than the simulated surface dipole mode.

2. We fully agree that silver and gold in themselves are in some respects better materials for plasmonics, e.g., because of weaker dampening and a smaller (Ag) or no (Au) oxide layer. That's why we, right from the start, pursued parallel efforts to measure the IMT at Al bars and short chains of Au nanospheres. We originally planned to include results from both samples into this proof-of-concept paper, but our analysis of the gold trimer chain showed no discernible IMT signal in the presence of heavy elastic vignetting, as seen in the elastic reference. Therefore, we concentrated our efforts, in correlating the experiment with simulations, on the aluminium, where the elastic vignetting was considerably weaker (which is not surprising considering the difference in scattering absorption). In the recent months we made further experiments on strongly scattering Au nanoparticles and refined our technique, but in our view, the results still do not permit an unambiguous interpretation in terms induced fields. With current instrumentation, the crucial elastic vignetting can be only removed by a full ptychographic reconstruction, while further instrumental improvements, permitting a high angular resolution also at small camera lengths and hence large convergence angles (i.e. smaller probe sizes), would allow to remove this effect even in simple IMT. A short overview on a representative measurement of a dimer of gold nanospheres (Fig. 1) is given below for further explanation.

Figure 1: High-resolution HAADF STEM image of a dimer of gold nanospheres (diameter 70nm) as used in the following measurements.

The inelastic EFDP of the gold dimer was acquired with a slit centered at 2.5eV (filtering a range from 2eV to 3eV), capturing the most prominent surface transverse mode (Fig. 2), which is homogeneously distributed over the surface.

Figure 2: Electron energy loss and IMT signals of a gold dimer.

Fig. 3(a) shows the integrated EFDP signal. Accordingly, we observe a large constant background from the C-foil (substrate) and a faint halo from the surface mode. Moreover, a strong damping of the signal on the nanoparticles (much stronger than in Al) is observed, which is clearly of elastic origin as demonstrated by the elastic reference signal 3(b). To remove these effects we subtracted the elastic reference signal (the normalization factor between inelastic and elastic signal has been determined in a region far away from the nanoparticle, in our case the left edge). The resulting background corrected inelastic signal in 2(c) now clearly shows the halo from the surface plasmon mode. In spite of the low signal-to-noise ratio in that measurement the observed experimental distribution of the loss probability corresponds approximately to the simulated one, where elastic scattering absorption has been taken into account (Fig. 3(a)). Similar to the Al rods the simulated values are approximately one order of magnitude larger than the experimental ones, which we also attribute to the role of the dielectric surrounding (i.e., the substrate and organic ligands).

Figure 3: Simulated loss signal and IMT for a dimer of gold nanospheres at a loss of 2.5eV. The signal has been convolved with the estimated beam width of 10nm.

Turning to the evaluation of the lateral momentum transfer the picture gets more complicated. Similar to the Al case we observe a similar magnitude of the experimental and simulated IMT. However, in the experimental data 2(d) a significant deflection only appears at the ends of the chain and not all around the structure, as expected from the simulation 3(b). We attribute this to the rather strong astigmatism (induced by the monochromator) in the STEM probe, which we could not prevent or compensate during the long acquisition times in the low-mag mode required for this sample. This astigmatism stretched the source approximately in the horizontal direction, where almost no IMT can be discerned. Noting that the probe momentum distribution is larger along the short axis of the STEM probe in position space and that deflection values of the elastic (not shown) and inelastic signal are about equal, we conclude that elastic vignetting cannot be excluded to be the dominant contributor in the inelastic IMT signal of the Au dimer. In summary, it certainly isn't lack of trying, why we did not include Au measurements in the paper but rather a combination of instrumental (mainly stability but also convergence angle) and principal difficulties with the gold sample (strong elastic vignetting). Notwithstanding our (hopefully relatable) reluctance, we would open to present the Au data in the Supplement if requested.

3. The admittedly large value for the background permittivity of 3.5 was determined iteratively by simulating energy-loss spectra until they matched the experimental one in the middle of the right bright spot of the mode. We ascribe this large background to the SiN substrate (e.g., [6]), Al₂O₃ shell and the additional C coating (added to suppress charging) as well as surface roughness effects [7], which we could not reasonably include in the simulation. In addition, we would like to raise the possibility that the lithographically fabricated Al rods exhibit a different dielectric response than bulk Al. We also observed that the use of a smaller background permittivity (e.g., 2.4) would only lead to a minor shift of the SP mode (in this case 0.2 eV), which approaches the error of our spectrum imaging measurement, and to a small change of the lateral momentum transfer (in this case several percent). Nevertheless,

we would like to stay with our criterion as we prefer the simplicity of it. We added a sentence in the Methods section for clarification on this question: “[...] *we increased the dielectric constant of the surrounding medium to 3.5 by aligning the mode in the experimental and simulated loss spectra (there was little change above 2.4 in the loss probability and the lateral momentum transfer).*”.

For the above reasons, I think that the present work is not mature enough yet, and I recommend against publication in Nature Communication. I believe, however, that this is an interesting technique with a lot of potential. If the authors could manage to perform additional experiments with plasmonic nanoparticles having more prominent SPR modes, ideally made of gold or silver, I would recommend reconsideration of the paper in Nature Communications, otherwise I suggest resubmission to a more specialized journal.

- Summing up, we have chosen the Al rods with some hindsight as we believe them to represent rather justifiable, even recommendable, proof-of-concept samples for the presented method. The application of this method to Au and Ag is experimentally analogous (and feasible as demonstrated) but more difficult to directly interpretate in terms of IMT, mainly due to the significantly larger elastic vignetting. We have included the Au data in this response to support this argument. Taking also into account that nanostructures made from lighter materials like Al or Cu became more popular for plasmonic applications (e.g., [8]), in particular in the UV range (in case of Al) we believe that our results are interesting in their own right. With our proof-of-concept we show that IMT permits a straightforward interpretation in terms of lateral electric and magnetic (not accessible with other methods) fields and provide the starting point for manifold future refinements, extensions and variants of this experiment. We elaborated further that there are no principal limits for IMT for samples of heavier elements but only an increased difficulty that can be overcome by further instrumental and experimental refinements as well as qualitatively new methods like ptychography. Consequently, we do not see why our chosen sample and the relative difficulty of applying this method to gold samples makes the methodic advance of less of a general interest insofar as to fall outside the scope of Nature Communications.

Reviewer #2 (Remarks to the Author):

The manuscript describes a very interesting experiment, in which both the electron energy loss (EEL) and the inelastic momentum transfer (IMT) of an electron beam passing an Al nanorod is shown. Specifically, the energy loss probability, a frequently investigated quantity, is combined in a novel way with the deflection of the electron beam, in order to obtain both longitudinal and transverse components of the electron’s self-induced field. While the spectrally resolved loss probability is a measure of the longitudinal induced field, the deflection measures the respective spatial Fourier component of the transverse field(s). In the analysis, a straightforward and well-justified theoretical description is given, with expressions linking the measured deflection with the position-dependent induced transverse field.

I believe this is the first experiment analyzing energy-filtered deflection maps for a resonant nanostructure. The experimental approach is innovative, the results are generally convincing and of good quality, and I generally believe the manuscript has potential for publication in Nature Communications. However, there are some fundamental issues concerning the presentation and placing into context of the manuscript,

which need careful attention by the authors to avoid misconceptions, before the manuscript could be further considered.

-Specifically (and I am not suggesting that they are doing so willingly), the authors appear to blur the lines between (1) a map of the induced field on the electron trajectory as a function of scan position near a nanostructure and (2) a map of a plasmonic mode of that nanostructure. In particular, several phrases in the text, the title, abstract and the quiver plot in Figures 3 and 4 appear to give that very impression. Indeed, one of the main claims of the manuscript is that it achieves a mapping of plasmonic fields, specifically plasmonic modes. Expressedly, the manuscript states: "In the following, we demonstrate the spectral field mapping of a surface plasmon mode of a lithographically produced Al nanorod, which was probed with an electron beam accelerated with 120 kV (see methods). For that, we selected the most prominent SPR mode, namely the dipole mode along the long axis of the Al rod." These are rather problematic claims, and from my perspective they are not sufficiently substantiated. These statements would imply to most readers that the field distributions obtained from the deflection maps, corresponding to the induced field, were closely related in direction and strength to the plasmon mode studied. However, it is evident that the induced field as a function of scan position will consistently point towards the nanostructure, which will always yield induced-field distributions with the same symmetry as the investigated structure, i.e., lacking a dipole contribution. This, of course, does not in any way reflect the symmetry of the plasmon dipole mode, which, as excitable by far-field radiation, is antisymmetric in nature. In other words, the technique is unable to obtain overall position-dependent phase information (unless Conversely, it is hard to see how a field component tangential to the structure (parallel to the rod axis) could be mapped, which is present in the plasmonic mode, but does not correspond to the structural symmetry. Overall, I am rather certain that readers from plasmonics will be confused when they see the monopolar (+quadrupolar) type of "field maps" in Figure 3.

- You very elegantly described a major challenge for all STEM EELS plasmon characterization, which is to provide a clear link to the optical response measured by optical far-field or near field techniques (e.g., SNOM). First of all, you are completely correct in stating that our language is misleading, when we say, we map the plasmon mode (if we can agree on referring to the singular surface modes (eigenmodes) as plasmon modes). We indeed map excitation probabilities or momentum transfers initiated by a focused STEM probe linked through modulated projection integrals to the oscillating surface charges and currents. Phase information with respect to the beam position is completely lost in this type of measurement. To remove this misunderstanding we now make clear at several points in the manuscript that we indeed map the response to point perturbations (e.g., at the end of the theory section "*As a consequence of the scanned probe only measuring fields it itself induced, both energy loss spectroscopy and IMT are not sensitive to the phases of the oscillating surface plasmon mode, if measured with an ideally focused STEM probe. Using an extended probe, the non-local response, including phase effects, affects the recorded signal.*" or at the beginning of the experimental section "*Note the positive sign of the loss probability of the dipole mode at both caps reflecting the previously discussed insensitivity toward phases of the oscillating SP within the STEM setup.*"). Nevertheless, we occasionally maintained the wording surface plasmon mode, because our measured losses and fields pertain to the dipole mode (without phase information). For instance, we just replaced "*we demonstrate the spectral field mapping of a surface plasmon mode*" with "*In the following, we demonstrate the spectral field mapping at a surface plasmon*

mode” (similar surgical modifications of the wording have been applied at other instances of ‘mode’). Indeed, in order to provide the full link between optical measurements (including, near field techniques such as SNOM) and EELS, we need to recover that phase information, which essentially boils down to determine the fully non-local dielectric response as provided by the inelastic ptychographic analysis of the full inelastic EFDP as described now in the revised supplement. To a certain extent experiments involving patterned illumination, such as in [9], can also provide certain projections of the non-local response. As written in response to the first referee, however, we believe that covering inelastic ptychography here in detail would be premature and leave the scope of the paper.

-The authors frequently use the term "transient field". For example, the caption in Fig. 3 calls these arrow plots "transient electrical field maps of the Al nanorod". While the field is certainly transient for an electron passing the nanostructure, I believe most readers will expect the measurement of "transient fields" as being related to some time-resolved measurement or a time-dependent phenomenon.

- We concur. Thus, we amended the caption of Fig. 3 and the second sentence of the Summary and Outlook to preempt this impression. The references to transient fields in the Abstract, Introduction and Theory are correct even if ‘transient’ is understood as temporally resolved. Also, we think that any references to our measurements are sufficiently qualified to pertain to a single mode, i.e. spectral component, only.

-The title "Spectral Field Mapping in Plasmonic Nanostructures with Nanometer Resolution" also implies that the field distributions shown relate to the plasmonic modes of the structure investigated. Also the term "Spectral Field Mapping" implies a spectrally dependent measurement, which appears possible for the future, but was not carried out.

- The term ‘spectral field mapping’ is used to account for the energy-filtering since no overall field is measured but rather the lateral components of a specific Fourier-component (as set by the energy selecting slit) of the field. Since the notion of an energy filtered electric field is strange we choose to refer to the spectral component instead. For the sake of clarity we changed the abstract to make clear that we measure only one spectral component.

In summary, these very nice measurements will be interesting to the plasmonic and electron microscopy communities in themselves, and I would suggest that less emphasis is placed on those claims the paper cannot satisfy.

Additional minor points:

-The caption in Fig. 2 does not denote the solid blue and dashed lines.

-The abstract states: "Notwithstanding, the direct imaging of transient fields permitting a direct mapping of plasmonic coupling, e.g., in terms of field enhancement, has been proven elusive to date. Here, we fill that gap..." There are notable other techniques, such as NSOM or PEEM, which provide phase-dependent field measurements in a manner at least as direct as what is shown here, I would say. This aspect is probably a little overstated in the manuscript, and corresponding references should be included.

- We completely agree and would like to express our regret over our, certainly not ill-willed, ignorance about other techniques. SNOM is capable of mapping the local optical density of states, whereas PEEM and cathodoluminescence map certain components of the induced fields. These techniques could also approach spatial resolution in the range of tens of nm, which is considerably larger than what could be achieved with a focused STEM probe and still larger than what we think is possible with our technique while still being able to resolve small deflection angles. However, they do not directly map the electric or magnetic fields as done with the IMT approach. We therefore only slightly modified the abstract to “*Notwithstanding, the direct and quantitative mapping of transient electric and magnetic fields characterizing the plasmonic coupling, e.g., in terms of field enhancement, has been proven elusive to date. Here, we demonstrate how to directly measure the deflection of a focused electron beam, which has excited a surface plasmon resonance, in a transmission electron microscope equipped with an energy filter.*” and included more references to other plasmon mapping techniques (including PEEM) further below in the text.

Reviewer #3 (Remarks to the Author):

The authors present a novel method for mapping the in-plane components of the electromagnetic field in plasmonic nanoparticles as a function of energy-transfer (i.e., for selected plasmonic modes). This is achieved using a form of energy-filtered differential phase contrast referred to as IMT in the text. Theoretical background, experimental evidence and conclusions are all provided in a clear and convincing way and are original. I strongly believe that this work will have a significant impact on the thinking within the field of electron microscopy and also inspire scientists in other fields. Therefore, I strongly recommend publishing it in Nature Communications, provided that the changes and comments listed below are taken into account.

== Content questions/comments ==

* In the Introduction, the authors state that TEM/EELS allow to map SPRs with nanometer resolution, yet in the theory and the Methods section, they give the beam diameter as 20 nm. In the Methods section, the authors also state that the step size was purposefully chosen much smaller than the beam diameter. Please clarify how all this affects spatial resolution (and what the purpose for choosing a small step size was).

- Although there is no optical limitation to focus the STEM beam even below one nanometer, we were forced to use a much larger diameter due to a number of other instrumental limitations. First of all, we had to resolve small deflection angles in the microradian regime, for which large camera lengths are required. To record the full diffraction patterns at such large camera lengths we had to reduce the convergence angle, which leads to an increased probe size. Moreover, the instability of the low-mag STEM setup (astigmatism drift, energy drift) imposed certain limits on the acceptable acquisition time for the whole EFDP (in the range of some tens of minutes). Because of that we binned detector pixels (facilitating faster read-out), which however, reduced the angular resolution further. There are instrumental solutions available to some of these problems (most notably fast CMOS electron detectors) but also stabilized monochromators and energy filters, however, they were not at our disposal). These and further anticipated optimization of the setup will likely reduce the beam diameter and hence the spatial resolution to the single digit nanometer range. We added a phrase stating that further reduction of the beam size is anticipated by employing advanced instrumentation such as direct pixel detectors.

- This is a misleading presentation on our part. Aware of the expected difficulties of vignetting and a large probe size (necessitated by the small convergence angle required to resolve small deflection angles in the diffraction pattern) we aimed from the beginning at a ptychographic evaluation of the data. There, such a small step size would allow the reversal of the information spread from the extended probe within the limits of the signal to noise ratio. This intent has no bearing on the IMT evaluation so we changed the methods section to reflect the nature of the intent (not just its existence).

* p. 3, left column, first paragraph: Ehrenfest's theorem has already been employed in [16] and its impact on STEM DPC has been discussed in more detail in Müller-Caspary et al., Ultramicroscopy 178 (2017) 62-80.

- We included the reference. Moreover, we now employ (generalized) Ehrenfest to justify the semiclassical approximation in a revised form of the supplement.

* Fig. 2: What are the dots in b) (COMs, maybe)? What are the two curves in c) (simulation and experiment? monochromated and unmonochromated?)

- The dots in b) are the CoMs and the two curves are the simulated and the experimental one. We updated the caption accordingly.

* In Fig. 2b, images 1 and 4 show non-uniform (and different!) intensity distributions, not just shifts. Where does this effect come from (it should not be vignetting as these positions are arguably far away from the interface) and how does it affect the analysis?

* In Experimental Results, the authors state that "the shift of the whole disk represents the inelastic momentum shift, whereas the redistribution of intensity within the disk may be also related to elastic and inelastic vignetting". In Data Treatment, they state that "To evaluate the IMT ... the CoM has been evaluated" after correcting for artifacts. Which method was used (CoM or disk shift)? According to [16] and Müller-Caspary et al., Ultramicroscopy 178 (2017) 62, the CoM should generally be used as it is related to the (average) deflection field by the Ehrenfest theorem. Naturally, how meaningful the average deflection field actually is will depend on the ratio between the characteristic length scales of the beam (diameter) and the field's variation, i.e. one will not be able to correctly measure fields varying on a length scale much smaller than the beam diameter. Constant fields lead to a simple shift of the diffraction disk, which can be evaluated by itself (e.g. by circle fitting) or by CoM - which should give the same results in this case. However, vignetting, partial absorption etc. will certainly "mess up" CoM calculations. They might not affect shift measurements as severely, but those are not directly interpretable in terms of the Ehrenfest theorem. Please clarify your procedure and how you overcame these issues (such as vignetting).

- For the sake of direct interpretability of the signal we used the CoMs, so the problems of vignetting and field gradients are present in our reconstructions. At the current state it is not

possible to correct vignetting for which a ptychographic reconstruction of both the elastic and inelastic signal would be needed (see revised Supplementary Information for more details).

* On p.3, right column, the authors state that they attribute the differences between the experiments and the simulations, among other things, to "the attenuation of the induced fields due to the oxide and carbon surface layers as well as the substrate". Were those (especially the substrate) included in the simulations at all?

* Fig. 3: The difference by a factor of ~ 10 between simulations and experiments in panels a) and b) is explained in the text (by an oxide layer and substrate effects, among other things). Why does the same effect not seem to occur in panels c) and d) for the in-plane components?

- We would like to answer the two preceding questions together. Given the general difficulty of correlating plasmon calculations to measurements in a quantitative manner it is hard to reason about these residual differences. The simulation uses a considerably simplified model of the specimen (no Al_2O_3 surface layer, no substrate, no C layer, idealized smooth surfaces and bulk dielectric function) offset by the heuristic model of a 'background permittivity' of the vacuum [10, 11]. Unfortunately, the MNPBEM option for including substrates does not work with the EELS mode as of today. A complete ab-initio modeling of the specimen is principally possible but very involved and would present original research in its own right. Nevertheless, the different deviations between theory and experiment in the loss-signal and the in-plane components provide a new perspective on the deficiencies of the theoretic model. For instance, it is conceivable that the substrate has a stronger effect on the loss than on the in-plane components of the E -field, as it significantly modifies the asymmetry of the out-of-plane component responsible for the loss. Further elaboration on the theoretic models as well as further experiments are underway to tackle this point.

* Fig. 4: the z component profiles seem to be asymmetric w.r.t. the particle. In particular, the profile in b) seems to have larger positive values (up to ~ 1.25) than negative values (down to ~ -0.75). In contrast, the profile in c) seems to have larger negative values (down to ~ -2.25) than positive ones (up to ~ 2). As the integrals over the profiles are related to the energy loss and the sign differs, does that mean there is energy gain in one of the cases? This might just be a drawing artifact, but should definitely be checked and corrected or mentioned and explained in the text.

- This is a misleading appearance in the plots, the larger amplitude of the negative peak in the z component of (c) is compensated by the slower decay of the positive amplitude above the sample. The integrals along these paths are both positive, the one of (b) being about 8 times larger than the one of (c).

* On p.5, right column, the authors mention the reconstruction of the optical density of states. This has been accomplished before using tomography, e.g., in Hörl et al., Nature Communications 8 (2017) 37.

- The photonic LDoS is only the spatial diagonal of the dyadic Green's function, whose reconstruction the tensor-field tomography approach would allow. So it is a remarkably reduced dataset, but we mention it as a link to existing methods and theory (the above citations has been included in the revised manuscript). Notably, the reported reconstructions always made use of some model based approach with considerable limitations (e.g. in the enforced sparsity in the energy-domain). Using IMT and tensor-field tomography would yield the LDoS directly.

* In the Methods section, please add some additional information, including the beam current, total number of beam positions, CCD exposure time per beam position, and number of pixels per CCD image. Also, how were the 170 μrad in STEM achieved?

- The parameters have been added to the B section of the Methods part. There we already outlined the strategy by which we achieved the small convergence angle: using a normally available low-mag mode. No further modifications of the microscope were required, although this is by no means a trivial mode to use combined with monochromation and energy-filtering in terms of stability but also beam calibration.

* In Sample Production, the authors mention that the "Al rods have been coated with a thin carbon layer to reduce charging". How was this done and how thick was the layer? In particular, was the substrate also coated, which would presumably result in a large reservoir of free charges and therefore alter or even destroy the surface resonances.

- The carbon layer was sputtered evenly over the sample to a thickness of about 5nm. Both the coating of the sample itself and the now conductive substrate are likely to have some influence on the plasmonic mode structure of the sample. Nevertheless, our imaged mode has clearly the spatial distribution of the dipole mode simulated without substrate but with the common effective medium approach. That suggests that the impact of the substrate mainly boils down to a red shift of the dipole mode considered here. Having said this, we believe that the substrate impact needs further elaboration in further plasmonic studies. Indeed, it has been realized that the substrate influences different energy regions in a different way (due to its varying dielectric function) and may eventually lead to the appearance of new resonances (e.g., due to ring currents). This problem can also not be avoided altogether, since any plasmonic structure needs to be supported (not only in the TEM holder). Therefore, the MNPBEM code employed in this study recently incorporated an option for layered dielectric surroundings, which, unfortunately, is not yet compatible with the EELS option (i.e., works only with optical excitations).

== Minor typos and formatting ==

* p. 1, last word in the left column should be "possess", not "posses"

* In the caption of fig. 2, "the beam positions ... indicated" should be "the beam positions ... are indicated"

* In the caption of fig. 4, "exhibit a the strong" should use either "a" or "the", but not both

* The references should be ordered (the introduction cites [4], followed by [2, 23], followed a bit later by [1])

* In ref [6], there is an encoding issue with one of the names

* Ref [16] lacks page numbers

- Thanks for pointing out these mistakes. They have been fixed.

References

- [1] F. J. García de Abajo. Optical excitations in electron microscopy. *Reviews of Modern Physics*, 82(1):209–275, feb 2010. ISSN 0034-6861. doi: 10.1103/revmodphys.82.209. URL <http://link.aps.org/doi/10.1103/RevModPhys.82.209>.
- [2] Knut Müller-Caspary, Florian F. Krause, Tim Grieb, Stefan Löffler, Marco Schowalter, Armand Béché, Vincent Galioit, Dennis Marquardt, Josef Zweck, Peter Schattschneider, Johan Verbeeck, and Andreas Rosenauer. Measurement of atomic electric fields and charge densities from average momentum transfers using scanning transmission electron microscopy. *Ultramicroscopy*, 178:62, 2017. ISSN 0304-3991. doi: <http://dx.doi.org/10.1016/j.ultramic.2016.05.004>. URL <http://www.sciencedirect.com/science/article/pii/S0304399116300596>.
- [3] A. Lubk and F. Röder. Phase-space foundations of electron holography. *Phys. Rev. A*, 92:033844, Sep 2015. doi: 10.1103/PhysRevA.92.033844. URL <http://link.aps.org/doi/10.1103/PhysRevA.92.033844>.
- [4] Andreas Trügler. *Optical properties of metallic nanoparticles*. Springer, 2011.
- [5] Kevin M. McPeak, Sriharsha V. Jayanti, Stephan J. P. Kress, Stefan Meyer, Stelio Iotti, Aurelio Rossinelli, and David J. Norris. Plasmonic films can easily be better: Rules and recipes. *ACS Photonics*, 2(3):326–333, 2015. doi: 10.1021/ph5004237. URL <http://dx.doi.org/10.1021/ph5004237>. PMID: 25950012.
- [6] Jürgen Waxenegger, Andreas Trügler, and Ulrich Hohenester. Plasmonics simulations with the mnpbem toolbox: Consideration of substrates and layer structures. *Computer Physics Communications*, 193:138–150, August 2015. ISSN 0010-4655. URL <http://www.sciencedirect.com/science/article/pii/S0010465515001228>.
- [7] Andreas Trügler, Jean-Claude Tinguely, Joachim R. Krenn, Andreas Hohenau, and Ulrich Hohenester. Influence of surface roughness on the optical properties of plasmonic nanoparticles. *Phys. Rev. B*, 83:081412, Feb 2011. doi: 10.1103/PhysRevB.83.081412. URL <https://link.aps.org/doi/10.1103/PhysRevB.83.081412>.
- [8] Mark W. Knight, Nicholas S. King, Lifei Liu, Henry O. Everitt, Peter Nordlander, and Naomi J. Halas. Aluminum for plasmonics. *ACS Nano*, 8(1):834–840, 2014. doi: DOI: 10.1021/nn405495q.
- [9] Giulio Guzzinati, Armand Béché, Hugo Lourenço-Martins, Jérôme Martin, Mathieu Kociak, and Jo Verbeeck. Probing the symmetry of the potential of localized surface plasmon resonances with phase-shaped electron beams. *Nature Communications*, 8:14999, April 2017. URL <http://dx.doi.org/10.1038/ncomms14999>.

- [10] Kristy C. Vernon, Alison M. Funston, Carolina Novo, Daniel E. Gómez, Paul Mulvaney, and Timothy J. Davis. Influence of particle–substrate interaction on localized plasmon resonances. *Nano Lett.*, 10(6):2080–2086, June 2010. ISSN 1530-6984. doi: 10.1021/nl100423z. URL <http://dx.doi.org/10.1021/nl100423z>.
- [11] Huanjun Chen, Feng Wang, Kun Li, Kat Choi Woo, Jianfang Wang, Quan Li, Ling-Dong Sun, Xixiang Zhang, Hai-Qing Lin, and Chun-Hua Yan. Plasmonic percolation: Plasmon-manifested dielectric-to-metal transition. *ACS Nano*, 6(8):7162–7171, August 2012. ISSN 1936-0851. doi: 10.1021/nn302220y. URL <http://dx.doi.org/10.1021/nn302220y>.

REVIEWERS' COMMENTS:

Reviewer #1 (Remarks to the Author):

I have carefully read the reply of the authors to my first report as well as the revised manuscript. The authors have worked hard to refute my criticism and have responded in full detail and very honestly. I truly appreciate these efforts.

However, I am still not convinced that the experimental results indeed show what the authors claim. I understand that this is a difficult experiment using a novel approach, so it is absolutely fine for me that a number of open questions remain. Nevertheless, as it stands the paper shows a single dataset for the electric field maps, and even the agreement with the simulation results is moderate (the measured and simulated plasmon energies do not match overly well). For a proof-of-principle experiment published in a prestigious journal I would have liked to see more evidence.

I again kindly ask the authors to provide additional results, concerning either the quadrupole peak of the nanorod or (probably more interesting) the field of coupled nanoparticles. It would be also fair to add the discussion about the difficulties with gold and silver nanoparticles in the main text. If the authors feel that the inclusion of novel experimental results is presently not possible, I suggest submission to a more specialized journal.

Reviewer #2 (Remarks to the Author):

I have thoroughly read the response by the authors, the rebuttal letter and the revised version of the manuscript. In my opinion, the authors have made an effort to clearly respond to all questions and criticisms, and they have carefully revised the manuscript accordingly.

I recommend publication of the manuscript in its present form.

Reviewer #3 (Remarks to the Author):

There are still a couple of "possess" that should read "posses" (which should be corrected in the proofing phase), but other than that I recommend publication.

Second Letter to the Referees for resubmission of NCOMMS-18-07042

August 17, 2018

We thank the referees again for their effort and comments.

On the request of referee #1 and the editor we included our measurements on the gold nanospheres in the supplement (with a reference in the main manuscript), including a detailed discussion of these results.

There we particularly highlighted the crucial role of elastic vignetting at strong elastic scatterers such as gold which prevent an interpretation of the IMT without more elaborate techniques to remove the vignetting, such as inelastic ptychography (the theory of which has been included in the supplement). On the other hand, the elastic vignetting in Al is considerably lower facilitating a quantitative interpretation of the IMT in terms of electric fields. We believe that this central result is well supported in the manuscript since we clearly show that the dipole mode is dominating within the filtered energy window (zeroth moment of energy filtered diffraction patterns shows the capping structure of the dipole mode) and the IMT (first order moments) corresponds remarkably well to the deflections obtained from modelling. We thusly disagree with the referee on the correspondence between the experimental and simulation data, which, in light of the general unavailability of sufficiently detailed models such complicated systems as the electron gas in non-ideal nanoparticles (e.g. surface roughness or non-homogeneous material composition). Indeed, truly quantitative low-loss EELS (particularly in terms of the loss probabilities) has not been established due to these complications in modelling, as a result loss probabilities are typically just given in arbitrary units. Several phenomenological approximations like the effective medium approach have been developed to account for some effects. Others like the impact of the surface roughness have been investigated in the literature without yielding practical approximation to take them into account in the simulation. We therefore consider the agreement of the IMT and the loss probability we obtained in our work rather remarkable.

The additional scenarios outlined by the referee (e.g., quadrupole mode or different nanoparticle geometries) are certainly interesting in themselves but would not give completely new insight into the technique. In particular the quadrupole mode would create additional experimental problems as the loss probability for this mode is greatly reduced with respect to the dipole one.

The grammatical errors mentioned by reviewer #3 have been fixed.

Reviewer #1 (Remarks to the Author):

I have carefully read the reply of the authors to my first report as well as the revised manuscript. The authors have worked hard to refute my criticism and have responded in full detail and very honestly. I truly appreciate these efforts.

However, I am still not convinced that the experimental results indeed show what the authors claim. I understand that this is a difficult experiment using a novel approach, so it is absolutely fine for me that a number of open questions remain.

Nevertheless, as it stands the paper shows a single dataset for the electric field maps, and even the agreement with the simulation results is moderate (the measured and simulated plasmon energies do not match overly well). For a proof-of-principle experiment published in a prestigious journal I would have liked to see more evidence.

I again kindly ask the authors to provide additional results, concerning either the quadrupole peak of the nanorod or (probably more interesting) the field of coupled nanoparticles. It would be also fair to add the discussion about the difficulties with gold and silver nanoparticles in the main text. If the authors feel that the inclusion of novel experimental results is presently not possible, I suggest submission to a more specialized journal.

Reviewer #2 (Remarks to the Author):

I have thoroughly read the response by the authors, the rebuttal letter and the revised version of the manuscript. In my opinion, the authors have made an effort to clearly respond to all questions and criticisms, and they have carefully revised the manuscript accordingly.

I recommend publication of the manuscript in its present form.

Reviewer #3 (Remarks to the Author):

There are still a couple of "possess" that should read "posses" (which should be corrected in the proofing phase), but other than that I recommend publication.